# Health Service Needs from a Household Perspective: An Empirical Study in Rural Empty Nest Families in Sinan and Dangyang, China

**DOI:** 10.3390/ijerph19020628

**Published:** 2022-01-06

**Authors:** Xueyan Cheng, Liang Zhang

**Affiliations:** 1Shanghai Health Development Research Center (Shanghai Medical Information Center), Shanghai 200032, China; xycheng6972@hust.edu.cn; 2School of Medicine and Health Management, Tongji Medical College, Huazhong University of Science and Technology, Wuhan 430030, China; 3School of Political Science and Public Administration, Wuhan University, Wuhan 430030, China

**Keywords:** health services need, household unit, empty nest household, collective model, empirical research

## Abstract

This study aimed to explore the health service needs of empty nest families from a household perspective. A multistage random sampling strategy was conducted to select 1606 individuals in 803 empty nest households in this study. A questionnaire was used to ask each individual about their health service needs in each household. The consistency rate was calculated based on their consistent answers to the questionnaire. We used a collective household model to analyze individuals’ public health service needs on the family level. According to the results, individuals’ consistency rates of health service needs in empty nest households, such as diagnosis and treatment service (H1), chronic disease management service (H2), telemedicine care (H3), physical examination service (H4), health education service (H5), mental healthcare (H6), and traditional Chinese medicine service (H7) were 40.30%, 89.13%, 98.85%, 58.93%, 57.95%, 72.84%, and 63.40%, respectively. Therefore, family-level health service needs could be studied from a family level. Health service needs of H1, H3, H4, H5, and H7 for individuals in empty nest households have significant correlations with each other (*r* = 0.404, 0.177, 0.286, 0.265, 0.220, *p* < 0.001). This will be helpful for health management in primary care in rural China; the concordance will alleviate the pressure of primary care and increase the effectiveness of doctor–patient communication. Health service needs in empty nest households who took individuals’ public needs as household needs (*n* = 746) included the H4 (43.3%) and H5 (24.9%) and were always with a male householder (94.0%) or at least one had chronic diseases (82.4%). Health service needs in empty nest households that considered one member’s needs as household needs (*n* = 46) included the H1 (56.5%), H4 (65.2%), H5 (63.0%), and H7 (45.7%), and the member would be the householder of the family (90.5%) or had a disease within two weeks (100.0%). In conclusion, family members’ roles and health status play an important role in health service needs in empty nest households. Additionally, physical examination and health education services are the two health services that are most needed by empty nest households, and are suitable for delivering within a household unit.

## 1. Introduction

The family plays an important role in the care and rehabilitation of individuals [1,2]. The family function is complex and focuses on the whole system rather than on the individuals, including roles (e.g., family structure), relationships, well-being, and belonging, and significantly takes place in the context of some public concerns, such as health problems [3,4]. In fact, the role of the family has been widely concerned in primary care in many countries. In the United States, family-centered care has incorporated into the standards and requirements in the performance system [5]. A Canadian study developed a Calgary Family Assessment and Intervention Model to evaluate family functioning in health research and family care [6]. Moreover, family functioning can influence health literacy among members. For example, children have e reported good health literacy within a better family context (better education, higher economic level, etc.) [7]. It is suggested that individuals in the same family could play their respective roles, achieve their practical goals, and maintain the relationships with each other; these are supportive factors to promote individual health [8,9]. In general, understanding the family experience of health and illness within the family’s social and cultural context helps health professionals provide targeted health services [10]. In the COVID-19 context, the pandemic poses an acute threat to the well-being of the whole family, which emphasizes the importance of family health [11,12].

Individuals’ health could affect other members’ health service needs in the same households. For example, for patients with chronic disease, their family caregivers take responsibility for maintaining treatment compliance, supervising medication intake, and providing emotional and economic support. However, family caregivers often show poorer health than those in families without a patient, and they are more likely to suffer from a host of problems, such as anger, fear, and depression [13,14]. People’s health is also associated with those with whom they are living. Support from the patients’ families, especially their spouse’s support, can affect their health-seeking behavior. For example, support from husbands plays an essential role in encouraging women’s health [15]. Additionally, because of the critical role of family members, some countries such as Canada, have tried to expand their efforts to actively involve patients’ family members in health service improvement and system redesign initiatives [16]. Hughes and Waite [17] found that married couples (age 51–61) living alone or with children are the most advantaged in health, but single women living with children are disadvantaged on all health outcomes. In China, old adults living alone and living with their children showed both advantages and disadvantages in health, while those with a spouse in the household provided the best health protection [18].

In China, the limited doctor–patient interactions are often confined to the brief consultation time and care facilities. However, family members’ behavior concordance and daily support could compensate for the situation [19], and family involvement could be critical for health management [20]. Therefore, many health services that could be delivered based on a household unit. In 2016, the Chinese Government proposed a family doctor system, in which general practitioners would establish a long-term service relationship with families that signed a contract with them. The family doctors would offer 95% of the family’s primary care and play a vital role as the primary health gatekeeper [21,22,23]. However, due to a lack of several essential uniform features such as health insurance support, appropriate incentive mechanisms, objective evaluation methods, and an effective way of delivering service, so there is some difficulty implementing the family doctor contracting services in China [24]. The family-centered care model has been promoted as a contemporary model of health service delivery, and evidence has shown that family-centered care for older adults is positive [25,26]. In the Chinese community-based healthcare setting, family-centered care showed a positive effect on seniors with diabetes [27]. Moreover, some health-promotion programs have tried to consider families, but due to a lack of funding and policy, it is difficult to maintain family-centered interests in associating interventions [28]. Therefore, it is crucial to clarify health service needs from a household perspective, which would be indicators for services, such as a family doctor contracting service.

In China, an empty nest household usually refers to households without children or whose children have left their parents’ home [29,30]. The accelerated urbanization and inequity of economic development in urban and country areas have resulted in the empty nest becoming the main family pattern in rural China. In 2016, empty nesters accounted for 51.1% of the elderly in China, and this proportion will reach 90% by 2030 according to China’s National Committee on Aging [31]. Often associated with low income, poor living conditions, and the lack of social and emotional support, residents in empty nest households are more likely to be vulnerable to different health problems and irreversible decreases in functional capacity [30,32,33]. Moreover, empty nest individuals are usually older adults who are likely to suffer from a high prevalence of chronic conditions and disability [34,35,36]. When their children move out of their homes, their empty nest parents are more likely to suffer from empty nest syndrome, resulting in loneliness, anxiety, frustration, etc. [37,38,39]. An investigation conducted in Sichuan, a western province in China, found that 30.11% of elderly empty nesters had anxiety-related symptoms [40,41]. These negative emotions are consistently tied to a subjective feeling of increased pain, disease, and tiredness [42,43,44]. In general, members in empty nest families are often concerned with poor health statuses, poor mobility, and a high risk of chronic diseases. Therefore, primary care facilities in rural areas are always under significant pressure in health service delivery, such as diagnosis and treatment, chronic disease management, physical examination, and mental health. Otherwise, with the advancement of the internet and social media, older adults have a high demand for remote assessment, such as telemedicine services, to help them become more independent in daily living activities [45,46]. Moreover, some empty nesters use complementary medicine, such as traditional Chinese medicine, to replace other kinds of treatment for economic or preference reasons [47].

Therefore, we assume that health services delivered based on a household unit should be more efficient, especially for empty nest households who have more health service needs but limited support. However, there is little concern about health service needs from a household perspective, the difficulty to collect sufficient family and individual information, and a lack of proper methods to analyze individuals’ health service needs from a household perspective. In this study, we conducted empirical research to determine the health service needs in empty nest households in Sinan and Dangyang in China from a household perspective.

## 2. Methods

### 2.1. Setting

This study was conducted in Sinan County in Guizhou Province and Dangyang County in Hubei Province, located in western and central China. Both counties are at the first-class economic level (GDP ranked 2/10, 4/13 in their cities in 2020). Sinan County has 7 townships, and Sinan County has 17 townships. Both countries are located in relatively flat areas, and the distribution of households is dense. Households in these counties are equipped with at least one village clinic, one township hospital, and county hospitals, and residents can access different health services from any health facility.

### 2.2. Study Design and Data Collection

With a 95% confidence level, the calculation is based on the requirement that the absolute sampling error does not exceed 3%. Due to the use of multistage complex sampling, the design effect will generally be between 2 and 2.5 [48,49]. This study considers the design effect at 2.5.
(1)$$n_{i}=u_{a}^{2}\cdot p\cdot (1-p)/\delta^{2}$$
(2)$$N_{i}=n_{i}\cdot DEFF$$

Equation (1) calculates the sample size of simple random sampling. $n_{i}$ represents the number of samples required for the *i* stage. The $u_{a}$ corresponds to the inspection level of *u* value, and δ is the allowable error. Equation (2) calculates the sample size of multistage stratified random cluster sampling. $N_{i}$ is the sample size of multistage stratified random cluster sampling. *DEFF* is sampling efficiency, referring to how many samples in this sampling process can provide the information that one sample could in a simple random sampling. The absolute sampling error is 21.338% of the chronic disease prevalence of the whole population based on the number of patients in China according to the 2013 National Health Service Survey, while the is 1.96 with a 95% confidence level [50]. Sampling efficiency could be affected by the intra-group correlation coefficient (ICC), sample stratification, the average number of respondents in each set, internal heterogeneity. Among them, the design efficiency of the National Census in China is 1.4, which serves as a reference value in this study. Therefore, the sample size in each county was 3584 individuals. According to the Fifth National Health Service Survey, the average amount of individuals per household was 2.9 [34]. Lastly, at least 1200 households in each county should be investigated. The sampling process could be found in Figure 1.

This study was conducted in Sinan County in Guizhou Province and Dangyang County in Hubei Province. The counties were selected with a purposive sampling strategy in central and western China. A multistage stratified random sampling strategy was used to select the households. Five townships in each county were selected randomly, and six villages were considered according to their distance away from each township, with two villages selected randomly far, medium distance, and near the central township. We conducted face-to-face interviews with around 40 households in each village, and each individual was questioned. Finally, we investigated 7293 individuals in 2735 households. The following criteria were used to select the empty nest households: (1) households with only two members in the house first entered the study; (2) households with two members must be spouses; (3) respondents must have lived at the survey site for at least six months; (4) at least one member in an empty nest household could participate in the study via face-to-face interview. Finally, 1606 empty nest people in 803 empty nest households were included (803 × 2 = 1606). Informed consent was obtained from all participants in this study.

### 2.3. Health Service Needs

According to Nobile [51], the WHO definition of health is a dynamic state of wellbeing characterized by a physical, mental, and social potential, satisfying the demands of life commensurate with age, culture, and personal responsibility. Accordingly, this study divided health service needs into physical health service needs and mental health service needs generally. In China, the health system is always concerned with healthcare and public health [52]. The healthcare system is designed to satisfy people’s medical needs, such as diagnosis and treatment and chronic disease management [50]. Public health is the collective action for sustained population-wide health improvement, such as health surveillance and preventive care [53,54]. With the advancement of social media, telemedicine has played an important role in providing health services in China, which should be considered when studying health service needs [55]. Therefore, we finally made the health service needs in this study as diagnosis and treatment services, chronic disease management service, telemedicine, physical examination service, health education service, mental healthcare, and traditional Chinese medicine service.

### 2.4. Health Service Needs Consistency Rates

According to individuals’ answers to the questionnaire, consistency rates of individuals’ health service needs could be calculated as:Consistency rate = COUNT *n* (*x*_1*ij*_ = *x*_2*ij*_, *i* = 1,2,……,*N*; *j* = 1,2,……,8)/*N*(3)

*x*_1*ij*_ refers to the health service need of one member in an empty nest household, *x*_2*ij*_ and refers to the other member’s health service need in the same empty nest household. When both selections of two individuals were the same, then this empty nest household would be counted into this study. *i* is the code of empty nest households, and *j* refers to different health service needs. *n* is the number when *x*_1*ij*_ = *x*_2*ij*_, and *N* is the total count of empty nest households in this study. If individuals’ health service needs in empty nest households do have consistency to some extent, then it is feasible to explore the health service needs from a household perspective.

### 2.5. Health Service Needs from a Household Perspective

#### Collective Household Model

Becker first illustrated the collective household model, in which the household is characterized as a collection of individuals. This model assumes that family consumption decisions result from multi-person decision making. An intrinsic feature of the collective model is the sharing rule, which governs the within-household distribution of household capitals. This sharing rule is always an indicator of the bargaining power of individual household members. The ultimate consumption decision on each good or service is always dependent on household characteristics, income levels, etc. However, these factors only affect different weights of individuals’ bargaining power in a household’s model but not the preferences of individual household members [56,57].

When there are two individuals (1 and 2) in the same household consuming a set of services, the health service needs in a household unit would be [57]:u(*Q,qf,qm*) = max(*Q,qf,qm*){*b*_1_
*uf*(*Q,qf,qm*) + *b*_2_
*um*(*Q,qf,qm*)}(4)

*Q* refers to the public health service needs and *qf* and *qm* are the private health service needs. *uf* and *um* are the expected health utilities people would obtain from different health services. In this study, we investigated people’s subjective health score, and the gap between their status to full-health status is the utility they would get from receiving health services. *b*_1_ and *b*_2_ refer to individuals’ bargaining power in the same empty nest household. It can be affected by individuals’ education level, role in their family (householder or not) and their objective health status valued by EQ-5D.

This study took the mean score of different factors as individuals’ bargaining power. If the educational level was divided into three levels, the higher level of an individual, the stronger their bargaining power. If individual 1 is a householder, then *b*_1_ = 1, *b*_2_ = 0. The value estimated by EQ-5D refers to the objective health status.

## 3. Results

### 3.1. Consistency Rates of Different Health Service Needs in Empty Nest Households

In this study, individuals’ consistency rates in diagnosis and treatment service, chronic disease management service, telemedicine care, physical examination service, health education service, mental healthcare, and traditional Chinese medicine service were 40.30%, 89.13%, 98.85%, 58.93%, 57.95%, 72.84%, and 63.40%, respectively. Therefore, family level health service needs could be studied based on this concordance.

### 3.2. Correlations between Individual’s Health Service Need in Empty Nest Households

In Table 1, health service needs, diagnosis and treatment service (*r* = 0.404), telemedicine care (*r* = 0.177), physical examination service (*r* = 0.286), health education service (*r* = 0.265), and traditional Chinese medicine service (*r* = 0.220) of individuals in an empty nest household have significant correlations with each other (*p* < 0.001).

### 3.3. Health Service Needs from a Household Perspective in Empty Nest Households

As shown in Table 2, individuals in empty nest households have similar subjective and objective health mean scores. Individuals with educational levels less than primary school accounted for over 50%.

As seen in Table 3, households with public health service needs as their household needs accounted for 93.4% (*n* = 746), and households that take one individual’s health service needs as their household needs accounted for 5.7% (*n* = 46). The public needs at the household level mainly include the physical examination service (43.3%) and health education service (24.9%). The individual needs mainly include the diagnosis and treatment service (56.5%), physical examination service (65.2%), health education service (63.0%), and traditional Chinese medical service (45.7%). In general, the health service needs in empty nest households mainly include the diagnosis and treatment service (12.4%), physical examination service (44.2%), health education service (26.9%), and traditional Chinese medicine service (18.9%).

### 3.4. Characteristics for Health Service Needs in a Household Unit

According to Table 4, the empty nest households who take public health service needs as household needs are mainly male householders, accounting for 94.0% (*n* = 803). Additionally, 69.3% of these households had at least one individual who had an illness within two weeks while we conducted the survey, and 82.4% of households had at least one individual with chronic disease. Therefore, households with at least one individual who had an illness within two weeks or chronic conditions were more likely to take public health needs as household needs.

Households that consider an individual’s health service needs as household needs are more concerned with the individual’s health. The individual usually had a disease within two weeks (100.0%) or was the householder in the family (90.5%). The results could be found in Figure 2.

## 4. Discussion

In this study, the health service needs of individuals in empty nest households have a high degree of consistency. This may be a credit to the shared family context they have, where their family culture and living habits originate, resulting in individuals in the same household having similar health literacy. For example, Wong et al. [58] found a health literacy information sharing system among family members; the health literacy among family members could be shared and could change individuals’ health behaviors. Ishikawa and Kiuchi [59] found that although an individual’s ability to achieve health-related literacy is limited, it could be compensated by other family members’ abilities. This may lead to the high degree of consistency in their health service needs.

According to the results of this study, the consistency rate was relatively low for diagnosis and treatment service needs. It is believed that people’s well-being and severity could be affected by their subjective feeling, so that people may have different health service needs, even though they have the same kind of disease [60]. However, service needs, such as physical examination, health education, and consecutive chronic disease prescription, are commonly considered as essential health service needs, and were added to the family doctor contract services [61], which are suitable to be delivered within a household unit.

In this study, individuals’ health service needs in empty nest households have been found to be positively correlated with each other. The environment and lifestyle that individuals share within a household result in similar health service needs [62]. Additionally, when one becomes sick, their spouse usually worries about catching the same illness, leading to a change in the individual’s health service needs. Therefore, family-oriented health promotion and disease prevention are promising strategies as family members may support and nurture one another through life stages [28]. In particular, empty nesters may live together with each other for a long time.

There are still difficulties faced in conducting family-centered health services in primary care. In this study, health service needs on a family level in empty nest households mainly include physical examination services, health education services, etc. People in empty nest households are almost always old, suffering from poor health status, and lacking economic support. For these people, only a basic level of health is required, and is hoped for to reduce the children’s burden [63]. In China, people over 65 years old could access free physical examination services, which showed positive outcomes in the timely discovery of health problems [64,65]. Health education can popularize common disease prevention knowledge, chronic disease prevention, daily maintenance, and acute disease measures for residents in rural areas with low educational levels. This kind of health education provided by primary care institutions is the main source for empty nest households’ health information [66]. However, these health services are not delivered from a family unit, although some are taken into family doctor contracting services [24]. The Chinese Government issued “Guiding Opinions on Regulating the Management of Family Doctor Contracting Services” in 2018, and it proposed that the number of residents signed to each family doctor should not exceed 2000, but there is more demand in reality. For example, the number of contract resident visits for each family doctor from 2013 to 2016 in Pudong, Shanghai, was more than 8000 per year [67].

Family members’ roles are essential in producing health service needs. In this study, residents in empty nest households may take householders’ health service needs as household needs, especially when the householder is male. This is also proven in other families. For example, Meydanlioglu et al. [68] found father’s educational status determined factors associated with their children being overweight and obese. Dongn et al. [69] mentioned that fathers with cognitive empathy could alleviate depression caused by mothers’ parental stress. Additionally, parents are critical in guiding children’s behavioral changes, such as food choice and physical activity [70,71]. Therefore, some health service needs could be delivered based on this concept. For example, it may be more efficient to provide health services, such as health education to householders in empty nest households, for which other family members would be affected.

Health service needs from a household perspective are dynamic, for the family environment is always changing via family relationships, interactions, beliefs, values, routines, and practices [71]. Any family member’s health change will lead to a change in a family’s health service needs. When someone has an emergency illness or severe disease, it is more likely that their health service needs should be satisfied first. In contrast, health service needs from a household perspective are also stable. When both individuals in the same households have similar health status, they may consider their common health service needs as their household needs [72], which is common among empty nest households.

According to the United Nations, the first goal of sustainable development is eradicating poverty and the divisive implications of its pathology [73]. Evidence has shown that “poverty due to illness” and “return to poverty due to illness” are the main cause of poverty in remote rural areas in China [74,75]. Empty nest families always have a risk of health-related poverty problems due to their fragility in economy and health. Making health service needs clear from the household perspective would be helpful to locate families with severe health problems that may result in poverty, which will improve sustainable development in rural China.

## 5. Conclusions

Families are vital to the health of individuals as they promote family members’ healthy choices and encourage health behavioral change. Family involvement in health services delivery has been taken into consideration in many countries. However, the relationship of individuals’ health services needs is less evident, and few studies have focused on the family-level health service needs. In this study, we took empty nest households in rural China as an example, and explored the health service needs from a household perspective based on a cross-sectional study. A wide range of samples were selected with a multi-level sampling strategy, and a family model was applied to determine family-level health service needs for empty nest households. According to the results, individuals’ health service needs in empty nest households are highly consistent and are positively correlated with each other, indicating that health service needs could be studied based on a household unit. It could be served as a reference for policymakers of primary care to improve the effectiveness of health management and reduce the pressure of primary care. It is also a promising strategy to promote doctor–patient communication with limited interaction. This will be helpful for health management in primary care in rural China, for the concordance will alleviate the pressure of primary care and increase the effectiveness of doctor–patient communication. Empty nest households who considered individuals’ public needs as household needs were usually with a male householder or were experiencing a chronic condition; the health service needs mainly included physical examination and health education. Health service needs in empty nest households who took one member’s needs as household needs included diagnosis and treatment, physical examination, health education, and traditional Chinese medicine services. The member was more likely to be the householder of the family or had a disease within two weeks. Therefore, family members’ roles and health status play an important role in health service needs in empty nest households. Additionally, family-level health services needs in empty nest households mainly included physical examination and health education, which could be indicators to deliver health services for a family unit. In addition, figuring out health service needs from a household perspective is helpful to locate health-related poverty families and improve sustainable development in rural China.

## Figures and Tables

**Figure 1 ijerph-19-00628-f001:**
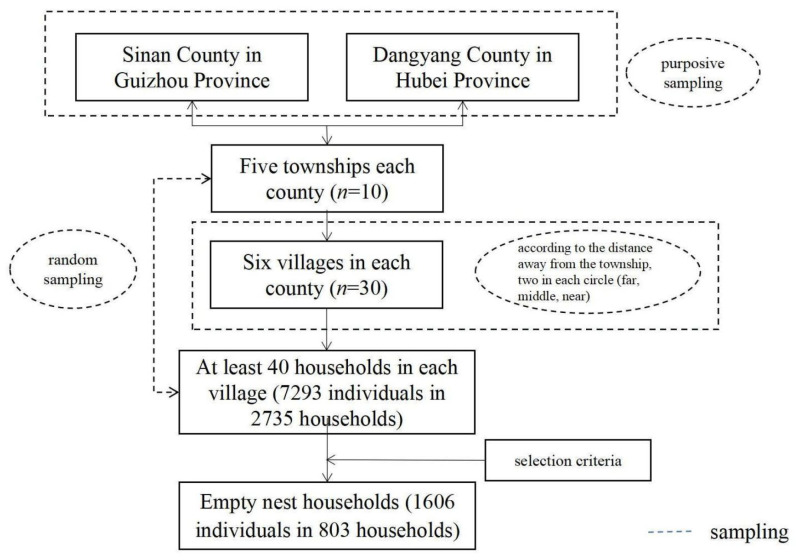
Selection process of empty nest households.

**Figure 2 ijerph-19-00628-f002:**
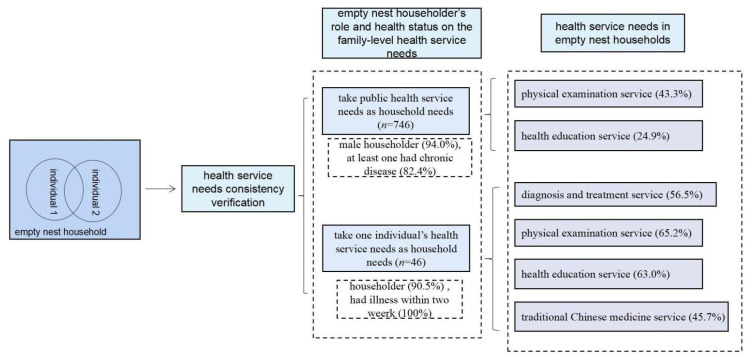
Health service needs in empty nest households.

**Table 1 ijerph-19-00628-t001:** Correlations between individuals’ health service needs (*r*).

Individual 1	H1	H2	H3	H4	H5	H6	H7
Individual 2							
H1	0.404 **	0.049	0.034	0.063	−0.044	−0.086	−0.045
H2	0.059	0.003	0.004	0.026	−0.036	0.006	0.025
H3	0.059	0.005	0.177 **	0.037	0.002	0.009	0.039
H4	0.003	0.033	−0.004	0.286 **	0.115 **	−0.013	0.015
H5	−0.032	−0.018	0.037	0.104 **	0.265 **	−0.018	0.051
H6	0.005	0.022	0.026	0.015	0.024	0.053	0.036
H7	0.007	0.027	0.031	−0.002	0.085 *	−0.035	0.220 **

Notes: H1: diagnosis and treatment service; H2: chronic disease management service; H3: telemedicine care; H4: physical examination service; H5: health education service; H6: mental healthcare; H7: traditional Chinese medicine service. * *p* < 0.05, ** *p* < 0.001.

**Table 2 ijerph-19-00628-t002:** Characteristics for each individual in empty nest households (*n* = 803).

Characteristics	Categories	Individual 1	Individual 2
Subjective health score	Mean score	69.95	67.69
Objective health score	Mean score	0.887	0.865
Educational level	Less than primary school	50.7%	60.0%
	Junior and senior high school	47.8%	36.6%
	More than undergraduate	0.3%	0.8%
Householder	Yes	74.5%	27.9%
	No	25.4%	71.8%

**Table 3 ijerph-19-00628-t003:** Health service needs in empty nest households (*n*, %).

Health Service Needs	Public Needs as Household Needs	Individual Needs as Household Needs			Total
		Individual 1	Individual 2	Total	
H1	73 (9.8)	11 (52.4)	15 (60.0)	26 (56.5)	99 (12.4)
H2	39 (5.2)	0 (0.0)	0 (0.0)	0 (0.0)	39 (4.9)
H3	1 (0.1)	1 (4.8)	0 (0.0)	1 (2.2)	2 (0.3)
H4	323 (43.3)	15 (71.4)	15 (60.0)	30 (65.2)	353 (44.2)
H5	186 (24.9)	16 (76.2)	13 (52.0)	29 (63.0)	215 (26.9)
H6	29 (3.9)	7 (33.3)	1 (4.0)	8 (17.4)	37 (4.6)
H7	130 (18.1)	11 (52.4)	10 (40.0)	21 (45.7)	151 (18.9)

Notes: H1: diagnosis and treatment service; H2: chronic disease management service; H3: telemedicine care; H4: physical examination service; H5: health education service; H6: mental healthcare; H7: traditional Chinese medicine service.

**Table 4 ijerph-19-00628-t004:** Characteristics for health service needs in a household unit in empty nest households (*n*, %).

		**Public Needs as Household Needs**				
			** *n* **		**%**	
Family characteristics	Householder’s gender	Male	701		94.0	
		Female	45		6.0	
	Two-week prevalence	both	154		20.7	
		One individual	335		47.7	
		None	236		31.7	
	Chronic disease	Both	228		30.7	
		One individual	384		51.7	
		None	131		17.6	
		**Individual Needs as Household Needs**				
			**Individual 1**		**Individual 2**	
			** *n* **	**%**	** *n* **	**%**
Individual characteristics	Householder	Yes	19	90.5	0	0.0
		No	2	9.5	25	100.0
	Two-week prevalence	Yes	21	100.0	25	100.0
		No	0	0.0	0	0.0
	Chronic disease	Yes	16	76.2	22	89.0
		No	5	23.8	3	12.0

## Data Availability

The data presented in this study are available on request from the corresponding author.

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
