# Peer review of "Health Service Needs from a Household Perspective: An Empirical Study in Rural Empty Nest Families in Sinan and Dangyang, China"

_ijerph, 2022, doi:10.3390/ijerph19020628_

Round 1

Reviewer 1 Report

The study analyse the consistency degree and people’s health service needs from the perspective of household through a multi-stage random sampling.

The approach of the work, although interesting from a methodological point of view in the analysis and in the sample, would deserve a greater bibliographic study in the introduction.

The results also appear undersized compared to the potential of the analysis. It is suggested to better frame the study through a systematic analysis of the literature with respect to the keywords identified.

The approach of the work, although interesting from a methodological point of view in the analysis and in the sample, would deserve a greater bibliographic study in the introduction.
The results also appear undersized compared to the potential of the analysis.
It is suggested to better frame the study through a systematic analysis of the literature with respect to the keywords identified.

A mapping review maps can be implemented thruogh the existing literature. Mapping reviews can be used to identify the need for yuor hypothesis.

In example in your introduction: "In 2014, the empty nests accounted for 51.1% of the elderly 86 in China, and this proportion will reach to 90% by 2030 according to the China’s National 87 Committee on Aging [27]. The increase in empty nest households has brought a series of 88 problems and thus has gained considerable attention from individuals, families, society, 89 and the entire state [28]. In general, members in empty nest families are often concerned 90 with poor health statuses because of lacking care from children, bad in mobility, and of 91 high risk in chronic diseases". Is this statement the only reference? it is a bit dated considering the importance of the research object you have chosen to analyze. In example, the link with telemedicine services has been completely omitted (digital divide and techonological perspective). 

At the same time I suggest, once these aspects of the literature have been identified with respect to your hypotheses, to align the final considerations that appear weak, since the general perimeter of reference is briefly described, but the type of contribution to suggest with respect to the result achieved.

Be more careful to use quotation marks "..." in stages such as for example:  ... According to WHO, health is defined as a dynamic state of wellbeing characterized 126 by a physical, mental and social potential, which satisfies the demands of a life commen surate with age, culture, and personal responsibility[29].  Is the sentence to be attributed to the WHO or to the bibliography indicated in the note? 

an other other similar cases in lines 279: In this study, health service needs in empty nest households are mainly concerned with health promotion and maintenance services (from sciencegate.app)

The work can be improved.

Reviewer 2 Report

Highlight changes in yellow in a next revision, please. No track changes.

Dear authors, MDPI does not require a structured abstract, please revise:

Abstract: Objective:”

Etc

Language needs revision: “

 This study was to analyse”

Revise spacing: “vice(H1)”

[All over : “rehabilitation[1,2].”

Or “Consistency rate(%)

Or “73(9.8)”]

Revise statistics: lower letter and italics: “P < 0.001”

Authors need to make the practical implications relevant… and clear: “

 It could be of great 32 help to improve the effectiveness of primary care and to focus more on households that need great 33 help on different health services.”

2. Methods

Where is the informed consent information?

Language needs revision… Was in?

“This study was in Sinan County in Guizhou Province and Dangyang County in Hu-102 bei Province, which are located in western and central China.”

Was selected… “which is selected”

Why start a new… paragraph with “but”:

“But the sample households were not all empty nest households.”

References needs to be here and all abbreviations defined at first use: “According to WHO,”

Any reference?

Not based in any previously known data?

Completely original, then explain the rationale, originality and novelty…

“According to individual’s answer to the questionnaire, consistency rates of individ-141 ual’s health service needs could be calculated as:”

And later o

When there are two individuals (1 and 2) in the same household consume a set of 160 services, the health service needs in a household unit would be: 161 u(Q,qf,qm) = max(Q,qf,qm){b1 uf(Q,qf,qm) + b2 um(Q,qf,qm)}

(2)

Address the italics in formulas? Or equations?

Italics… “Table 1. consistency rates of different health service needs (n = 803).” And start by upper letter

Any tables lack clarifying headings above specific content (empty cells above)…

Family characteristics

Householder’s gender

4. Discussion

Outside introduction, the incusing of references must be clear, thus normally usually direct style citations, including authors names, otherwise why is that particular reference there, to explain what?

“health literacy and consumption value[37, 38].”

This is based in what results? Clarify: “According to the results of this study, individuals in empty nest households had high 226 consistency degree in health promotion and maintenance services needs.”

A discussion like this must be supported in references, rather scarce in this section

Due to the nature of this text…

It would be important to make the connection to The United Nations Sustainable Development Goals: SDGs..

“Therefore, improving health status in 272 rural area has become an important step to get rid of poverty.”

I believe authors would necessarily need to address it...

Use plural here: “5. Conclusion

Start by a brief contextualization to defend your study…

5. Conclusion 276

Individuals in empty nest households have common health services needs with each 277 other, and one individual’s health service needs could affect others.”

Conclusions need to be entirely redone

See that along with the title and abstract, they are crucial in translating the relevance and content of the manuscript.

Thy need to include: brief contextualization and methodology, ten main findings and practical implications.

I do not find the conclusions suitable, at all…

I do believe authors failed in exploring the results. In my perspective tables are not enough, existing the need to add enlightening and updated graphics with results obtained, limiting the non-appealing tables to a minimum.

More reverences from 2021 should be included…

Round 2

Reviewer 2 Report

Highlight changes in yellow in a next revision, please. No track changes.

The non-published material says nothing

The Editorial services will check

 Answer: All the participants in this study were asked to sign an informed consent form. The consent form is shown below.”

No print screen or proof that this has actually been used

I am sorry to say that English is not acceptable, just see:

 According to the results, individuals’ consistency 18 rates in health service needs in empty nest households, such as diagnosis and treatment service 19 (H1), chronic disease management service (H2), telemedicine care (H3), physical examination ser-20 vice (H4), health education service (H5), mental healthcare (H6), and traditional Chinese medicine 21 service (H7) were 40.30%, 89.13%, 98.85%, 58.93%, 57.95%, 72.84%, and 63.40%, respectively.”

The sentence has no conclusion…

For that alone, the manuscript is not acceptable, see:

“ According to the result of 24 family collective model, health service needs in empty nest households in this study from a house-25 hold perspective mainly include H1 (12.4%), H4 (44.2%), H5 (26.9%), and H7 (18.9%).”

When the English is not acceptable the manuscript cannot be reviewed.

I must be clear.

Since I was extensive before, I am just clarifying this point…

Spacing note corrected:

p< 0.001).”

Reference number needs to be next to authors names, when directly cited…

As done everywhere

No formulas but equations:

“Formula (1) calculates”

Conclusions:

Why “tried”?

“and tried to explore”

Why upper letter? “And family”

I find the conclusions too general and unclear in findings

Terrible figure (quality) and unclear links (separate shapes):

Figure 1. Selection process of empty nest households”

Figure 2. Health service needs in empty nest households”
